# Morphologic Mandibular Bone Changes on Panoramic Radiographs of Children and Adolescents with Congenital Heart Disease

**DOI:** 10.3390/children10020227

**Published:** 2023-01-27

**Authors:** Fatma Saraç, Fatih Şengül, Periş Çelikel, İbrahim Saraç, Aybike Baş, Sera Şimşek Derelioğlu

**Affiliations:** 1Department of Pedodontics, Faculty of Dentistry, Ataturk University, 25240 Yakutiye, Turkey; 2Department of Cardiology, Erzurum Education and Research Hospital, University of Health Sciences, 25240 Erzurum, Turkey

**Keywords:** congenital heart disease, fractal analysis method, mandibular bone density, radiomorphometric measurement

## Abstract

Congenital heart disease (CHD) has effects on growth and development. However, information on how the structure of the mandibular bone is affected is limited. In the present study, we aim to compare mandibular bone structures of children affected with CHD and healthy ones through the fractal analysis method and radiomorphometric indices based on panoramic radiographs. The study consisted of 80 children (20 with cyanotic CHD, 20 with acyanotic CHD, 40 control) who were diagnosed with CHD and were treated through interventional therapy or followed up through medical therapy. Fractal dimension (FD) was performed in three different areas (angulus, corpus, and interdental bone) on 80 panoramic radiographs. Additionally, we assessed various radiomorphometric indices: mandibular cortical width (MCW), panoramic mandibular index (PMI), mandibular cortical index (MCI), and simple visual estimation (SVE). *p* < 0.05 was accepted as statistically significant in the analysis. Values of mean MCW, PMI, MCI, SVE, and FD measurements in children affected with CHD were found to be similar to the control group, regardless of whether they were cyanotic or acyanotic (*p* > 0.05). In this study, fractal analysis and radiomorphometric indices revealed no trabecular structure and mineral density changes in mandibular bone of children and adolescents with CHD compared to healthy subjects.

## 1. Introduction

Congenital heart disease (CHD), a structural abnormality of the heart or intrathoracic great vessels occurring during the fetal development is the most widespread congenital pathology and the major death cause among the children with congenital malformations [1]. It occurs in approximately 0.8–0.12% of all live births [2]. CHD is subdivided into two groups as acyanotic and cyanotic CHD. Cyanotic CHD causes right–left shunt flow leading desaturated blood to mix with the systemic arterial circulation. Although the tissues are adequately perfused in the patients with cyanotic CHD, hypoxia occurs when the oxygen level required by these tissues falls below the physiological level. Regulations at the gene level of this pathophysiological condition include many changes in metabolism, including neovascularization, increase in red blood cell production, and anaerobic glycolysis [3,4]. Long-term exposure to hypoxia may affect all organs and the body, including the jawbone. Especially, growth and development of cyanotic CHD patients are expected to be impacted [5]. 

Panoramic radiographs are the most preferred imaging method as a simple, practical, and inexpensive imaging method for the examination of the jawbones of the patients who apply to the dentist for any reason. In dentistry, panoramic imaging can be used for different purposes together with the assessment of growth and development. Bone mineral density loss can be measured with panoramic radiographic images [6]. Densitometric and radiomorphometric measurement methods have been used in many studies to evaluate bone qualitatively and quantitatively. In order to determine bone mineral density, various panoramic radiomorphometric indices such as mandibular cortical width (MCW), panoramic mandibular index (PMI), mandibular cortical index (MCI), and simple visual estimation (SVE) have been used in the studies [6,7,8,9]. Bone tissue has a complex structure, and researchers have developed different methods to analyze this tissue. However, in recent studies, it has been stated that density measurements or panoramic indexes that evaluate the macro-structure of the bone are not solely sufficient for the correct analysis of bone tissue; instead, the micro-architecture of the trabecular bone is extremely important in this regard [10]. Fractal analysis, which has been commonly used in both medicine and dentistry over the past few years, is a numerical texture analysis method based on a series of operations called fractal mathematics to describe complex shapes and structures in bone [11].

Bone tissue has a structure containing many small components, and researchers have developed many methods to study this tissue statistically and structurally. Fractal analysis, on the other hand, is a numerical tissue analysis method that is widely used in medicine and dentistry in recent years, based on a series of operations called fractal mathematics to describe the complex morphological and structural patterns of the bone. It is stated that the fractal dimension (FD) observed on the radiographic images shows the variations in trabecular bone density and bone mineral loss [12]. The complexity of the analyzed structure increases in line with the increase of FD. The box-counting method submitted by Rudolph et al. [13] is one of the most widely used fractal dimension analyses to assess the morphology of trabecular bone and spaces. A form of boxes, 2 to 64 pixels in size, is produced based on the radiographic image. FD was calculated from the slope of the line fitted on the logarithmic total box count-box size graph.

Muscles and bones of the children affected with CHD are subjected to many harmful effects during the growth, whereas it is well known that adults with advanced cardiac pathology are burdened with low bone mineral density [14]. CHD-affected children and adolescents with Fontan circulation have exhibited lower bone mineral densities and delayed bone age [15]. In addition to affecting growth and development, CHD also has effects on oral and dental health. Compared to the healthy children, children with CHD have higher susceptibility to dental caries [16], developmental defects of enamel [17], gingival inflammation, and periodontal diseases [18], whereas their saliva-buffering capacity and salivary pH [19] are lower. It has been reported that primary teeth in the children with CHD have irregularly oriented enamel prisms and lower Ca and P content [17]. To the best of our knowledge, the related literature indicates no study investigating the effects of CHD on mandibular bone using FD and panoramic radiomorphometric indices in children and adolescents. Dental practitioners should be extremely careful when invasive dental procedures such as tooth extraction or traumatic injury management are performed in patients with low bone mineral density for avoiding treatment complications. In the present study, we aimed to assess cortical and trabecular mandibular bone morphology of the CHD-impacted children and adolescents comparing them with a healthy group, using fractal analysis and orthopantomogram (OPG) based various panoramic radiomorphometric indices. The null hypothesis proclaims that there is no difference in the evaluation of cortical and trabecular mandibular bone morphology made with fractal analysis and four radiomorphometric indices in the children with CHD and healthy ones.

## 2. Materials and Methods

This retrospective study was carried out in Faculty of Dentistry’s Department of Pedodontics at Atatürk University, and an approval was obtained from the Atatürk University Faculty of Medicine’s Ethics Committee (dated as 28 April 2022 and Approval #19). The study was conducted in accordance with the Helsinki Declaration and parents’ written informed consents were collected. 

### 2.1. Inclusion Criteria

The study group included 40 children who were regularly examined in the Faculty of Medicine’s Pediatric Cardiology Department at Atatürk University and were previously diagnosed with CHD by cardiac evaluation and imaging (transthoracic echocardiography performed by an experienced cardiologist for all patients, transesophageal echocardiography, thorax CT, cardiac MRI, and right heart catheterization), received interventional treatment, or were followed up through medical treatment. Twenty of the children in the study group had a previous cyanotic CHD history, and the remaining twenty had a history of acyanotic CHD. The control group included 40 healthy children without any systemic disease who visited to our clinics at the Department of Pedodontics for their regular dental treatments and whose OPGs were also taken for radiological evaluations. New X-ray scans was not requested to prevent all patients from being exposed to more radiation, and radiographs were accessed from the hospital archives.

In order to avoid age and gender differences between the groups, patients with the same age and gender, like those included in the cyanotic patient group with the lowest incidence, were selected for the acyanotic and control groups.

### 2.2. Exclusion Criteria

The patients with metabolic bone disorders, thyroid problems, disorders of adrenal glands, systemic diseases, cirrhosis, chronic renal failures, gonadal pathology, vitamin D, parathormone and calcium metabolism disorders, patients with a previous or present use of medications effective on bone tissue, and patients with skeletal Class II or III malocclusions were excluded from the study. In OPGs in which the mental foramen and the regions of interest (ROIs) for fractal analysis cannot be clearly observed were also excepted.

### 2.3. Data Collection

In addition to routine cardiac examinations of children with CHD, peripheral oxygen saturations (SpO_2_) of the patients were measured from the fingertips via a pulse oximeter (Nellcor^TM^ Bedside SpO_2_ Patient Monitoring System, Medtronic, Minneapolis, MN, USA), and cyanosis was described when SpO_2_ was ≤92%. Demographic data and clinical information were obtained during routine examination. In addition, historical records and laboratory data were obtained from the hospital database. The congenital heart diseases guideline was regarded as reference for the diagnosis, classification, and management of CHD in children [20].

### 2.4. Radiomorphometric Indices

MCW: average width of the mandibular cortical bone (Figure 1(a)) measured through the perpendicular line from mental foramen to the lower margins of mandibula at both sides [21].

PMI: Average ratio of width of the mandibular cortical bone (Figure 1(a)) to the distance between the lower edges of mental foramen and the lower margins of mandibula (Figure 1(b)) bilaterally calculated [22].

MCI: Klemetti et al. [23] morphologically categorized the mandibular cortical bone in 3 classes:

C1: Normal cortex; flat and sharp bilateral endosteal margins.

C2: Moderately worn cortex; endosteal margin with semilunar (lacunar resorption) and/or endosteal cortical residue-like defects on both sides.

C3: Over-eroded/porous cortex; with extremely porous/eroded cortical layer including heavy endosteal cortical residues.

SVE: Mandibular inferior cortex classified as thin or not thin based on the observer’s experience [7].

### 2.5. Fractal Dimension (FD) Analysis of Panoramic Radiographs

All digital radiographs were taken with the same panoramic device (ProMax^®^, Planmeca Oy, 00880 Helsinki, Finland). The exposure settings were set on average as 65 kVp, 5 mA and 16.2 s. The patients were positioned in accordance with the manufacturer’s instructions by adjusting their Frankfurt planes parallel to the ground and their sagittal planes to the vertical red laser line determined of the OPG machine. All panoramic radiographs were recorded in DICOM format. Each pixel size equated to 28.4 μm.

All panoramic radiographic measurements were performed by a single trained observer (FS), using the ImageJ 1.3 (National Institutes of Health, Bethesda, MD, USA). The films were scaled 4 times in the x and y planes using bilinear interpolation. The region of interest (ROI) (240 × 240 pixels square) for each patient was standardized for each patient. For the measurements on the panoramic films, 3 different areas were determined: from the geometric center of the right mandibular condyle (ROI1), just below the mandibular canal in the right mandibular angulus region (ROI2), and from the apical level of the right teeth 5 and 6 (ROI3). Anatomical formations such as mandibular canal, cortical bone, lamina dura, and periodontal space from the ROI areas were not included (Figure 1). 

A modified version of the method used by White and Rudolph [13] for fractal analysis in analog panoramic radiographs was used for use in digital panoramic radiographs. ROIs in each image were selected and duplicated twice. Gaussian filter (sigma = 10) was used in second duplications in order to eliminate fluctuations in the brightness associated with the superimposing of soft and different bone tissues. The blurred image was removed from the first duplicate, which has the original image. The color depth of the image, which was 16 bits in lossless DICOM format, was reduced to 8 bits and converted to tiff format. In order to isolate bone marrow cavities and trabecula, RGB color 128 (gray) was added and the image was then converted into binary with a thresholding value of 128; thus, trabecula and marrow spaces were made possible to be more easily recognized. Image noise was eliminated by eroding and dilating. The image was then inverted and skeletonized and superposed on the original image, and the conformity of the skeletal structure to the trabecular structure in the original image was observed (Figure 2). Box-counting method was used for measuring FD of the skeletonized image (Figure 3). 

FD comparisons were first fulfilled between children with CHD (*n* = 40) and healthy children (*n* = 40). Then, children with both cyanotic (*n* = 20) and acyanotic (*n* = 20) CHD were compared with healthy children (*n* = 20). Finally, fractal dimension analysis was performed among cyanotic children with low oxygen saturation (*n* = 11) and acyanotic children with normal oxygen saturation (*n* = 11) and healthy children (*n* = 11).

To access inter-observer variability of MCW, PMI calculations, MCI, SVE, and fractal analyses were re-made on 25 patients after 3 weeks.

### 2.6. Statistical Analysis

Based on the Yagmur et al. study [7] on FD and radiometric indices, the number of patients included in the present study was adequate to obtain a power of at least 95% with an effect size of 1.2. Analyses were based on standard unpaired *t*-tests, with a significance of α = 0.05.

In the statistical analysis of the study, SPSS 26.0 (IBM SPSS Inc., Chicago, IL, USA) was used. Levene’s test was used to assess homogeneity of variances and normal distribution. For parametric data Student’s t-test and one-way ANOVA was used to compare quantitative variables between the groups and for non-parametric data Kruskal–Wallis and Mann–Whitney U tests were used. MCI and SVE distributions in the groups were compared by Chi-square test. interclass correlation coefficient (ICC) and kappa coefficient were used to assess inter observer reliability. The level of significance was determined as *p* < 0.05.

## 3. Results

The study consisted of a total of 80 children, including 40 children with cyanotic (6 girls, 14 boys) and acyanotic (6 girls, 14 boys) CHD and 40 healthy (12 girls, 28 boys) children. Children aged 5–18 years old were included in this study. While the mean age of healthy children was 10.2 ± 3.9, the children with heart disease were observed to have a mean age of 10.1 ± 3.9 (*p* = 0.886). The mean saturation values for cyanotic CHD were determined as 87.84 ± 10.56.

The ICC indicated an excellent reliability for MCW, MCI, SVE, ROI1, and ROI2 (kappa coefficients varied between 0.92 and 0.97) and good reliability for PMI and ROI3 (kappa coefficients were 0.86 and 0.88, respectively). Mean MCW and PMI measurements in children with CHD were similar to the control group (*p_MCW_* = 0.367 and *p_PMI_* = 0.5, Table 1). Additionally, MCW and PMI measurements in cyanotic or acyanotic children were similar to the control group, too (*p_MCW_* = 0.844 and *p_PMI_* = 0.317). The MCW value had a correlation with the age (*p* < 0.001).

The results of MCI and SVE distributions are given in Table 2. The distributions of MCI and SVE scores of children with CHD were similar to the control group (*p_MCI_* = 0.222 and *p_SVE_* = 0.312). A similar situation was observed between the control group and those with cyanotic or acyanotic CHD (*p_MCI_* = 0.083 and *p_SVE_* = 0.131).

In CHD, the SVE value had a correlation with the mean MCW value (*p* = 0.039) but showed no correlation with PMI (*p* = 0.319). On the other hand, there is no significant correlation between MCI value and the mean MCW and PMI values (*p* = 0.243 and *p* = 0.167, respectively). In the “thin” group, mean MCW values were found to be as 3.09 ± 0.65 mm and in the “not thin” group they were calculated as 3.69 ± 0.88 mm, respectively, and the mean PMI values were 0.4 ± 0.09 mm and 0.42 ± 0.12 mm in the “thin” and “not thin” groups, respectively.

In the control group, SVE value correlated with the mean MCW value (*p* = 0.002) and PMI (*p* = 0.022). However, MCI value did not correlate with the mean MCW and PMI (*p* = 0.163 and *p* = 0.215, respectively). In the “thin” and “not thin” groups, the mean MCW values were found to be as 2.54 ± 0.46 mm and 3.66 ± 0.85 mm, respectively, whereas the mean PMI values were 0.37 ± 0.08 mm and 0.44 ± 0.08 mm.

The FD values of the three regions of the mandibular bone in children with CHD were similar to the control group (*p* = 0.53). Additionally, no statistical difference was observed between the control group and the children with cyanotic or acyanotic CHD (*p* = 0.639, Table 1, Figure 4 and Figure 5).

## 4. Discussion

Panoramic radiomorphometric indices and FD have been used to determine the effects of many systemic diseases on the mandibula [6,24,25]. To the best of our knowledge, the studies regarding the morphological variations in the mandibulas of the children and adolescents with CHD seem to be unavailable in the literature. The present study (depending on whether the children with CHD were cyanotic or acyanotic) was conducted to determine the changes in the mandibulas of the children and adolescents with CHD through FA and dental panoramic radiomorphometric indices (MCW, PMI, MCI, SVE), and it revealed no statistically significant difference regarding trabecular structure and density of the mandibular bone.

Having a complex pathophysiology, bone development is affected by gender, age, hormones, diet, and lifestyle [26]. In their study with 75 children aged 0–6 years using quantitative ultrasound, Laura Gabriela et al. [27] reported that bone quality was lower in children with CHD compared to the healthy children, regardless of nutritional status. In their study using the DEXA method, Truong et al. [28] concluded that 1/3 of adults with CHD had lower values than the reference bone mineral density values in the same region. However, they emphasized that this situation is more stable in the younger patient group and that it is possible to associate the increased risk factors with osteoporosis in adults affected with CHD. In their study to investigate the mineral density of different bone regions, Elsharkawy et al. [5] argued that out of 39 children with cyanotic CHD, 6 children had low bone mineral density values, 13 borderline values, and 20 normal values and reported that children with cyanotic CHD may have a lower bone mineral density compared to healthy children. The null hypothesis was accepted in the present study because there was no difference in the OPGs evaluated with the fractal analysis method and radiomorphometric indices, regardless of whether children with CHD are cyanotic or acyanotic. The lower mean saturation values that were obtained in the study of Elsharkawy et al. [5] (74 ± 6.4) compared to this study (87.84 ± 10.56) may have caused this difference. However, similarity of the FD and PMI values in both groups might be due to the inclusion of patients who were previously diagnosed with cyanotic CHD and then treated in the cyanotic CHD group in this study; the limited number of patients; and the wide age range. In addition, early diagnosis and treatment of CHD has been possible with the developments in diagnosis and treatment methods in recent years. We think that this may be related to the similar FA and radiomorphometric index results in both groups of our study.

In previous studies conducted on patients with ß-thalassemia [7], type 1 diabetes mellitus [29], hemoglobinopathielia [8], and osteoporotic bone structure, MCW and PMI values of the patients were found to be lower than the healthy control groups. The present study is different from previous studies by suggesting that both groups had similar MCW and PMI values. This result may suggest that the aforementioned diseases have more negative effects on bone mineral density compared to CHD.

To the best of our knowledge, in the literature, there is no study evaluating the cortical bone thickness of patients with CHD via the SVE index. The studies conducted on the osteogenesis imperfecta-affected children indicated a relationship between the MCW and the SVE [25]. The mean MCW values suggested by Apolinaìrio et al. [25] ranged between 2.5–2.8 mm in the “thin” and 3.5–3.7 mm in the “not thin” group. In our study, the mean MCW values for CHD were observed as 3.09 ± 0.65 mm in the “thin” and 3.69 ± 0.88 mm in the “not thin” groups. In the present study, higher mean MCW scores than those of Apoloniaìrio et al. [25] were obtained. This can be explained by the fact that osteogenesis imperfecta affects the bone structure more than CHD. Additionally, the authors believe that the correlation between MCW and age has an impact on the similarity of MCW values in terms of SVE values.

It has been reported that adults with complex CHD have low BMD and osteoporosis risks continue even after the risk factors have been eliminated [30]. Although periodontitis is a common oral pathology mostly of bacterial origin, aforementioned osteoporosis or decreased bone mineral density may be a risk factor for alveolar bone loss in periodontitis [31]. Periodontitis is an inflammatory disease in which bone loss is irreversible and results in tooth loss following tooth mobility. In histomorphometric and microradiographic studies, it has been shown that the porosity of the mandibular cortical bone increases with the decrease in BMD [32]. Furthermore, when bacteria playing a role in periodontitis are present in the body as a source of focal infection, they cause health problems in patients with heart disease by creating a major risk factor [33]. In light of this information, it may be said that this vicious circle in heart disease negatively affects the health of individuals with CHD. It has also been reported that the risk of infective endocarditis is higher in children with CHD than healthy children [34]. In the present study, no significant difference was observed between the individuals with CHD and healthy ones regarding the radiomorphometric indices. However, different results may be encountered when a similar study is conducted in the adults.

The limitations of this retrospective study may be the lack of the occlusal situations that can be affect the mandibular bone morphology and the non-use of DEXA, which is a golden standard in the evaluation of bone mineral density. However, it can contribute to the literature since currently there are no studies assessing the mandibular bone density in the children and adolescents with CHD.

The study samples also included patients who had major operations in the early period into children in the cyanotic CHD group, which can be observed in a small number of patients with the diagnosis and treatment methods that are still under progress in our country. It may have affected the results of the study as the patients who had major operations in the early period were less exposed to the effects of CHD. In the current study, no evaluation was realized regarding some possible factors for low mandibular bone density, such as physical activity restrictions and other biomarkers that are common in the children affected with CHD. Although this study indicated no difference between the children with CHD and healthy ones in the trabecular mandible morphology, it was recommended to cover these candidate factors in the future studies. It is expected to follow the same patients and compare them with their peers in adulthood in further studies.

There are some recently conducted studies assessing the bone morphologies of the patients with various systemic diseases with different methods of radiological analysis. However, there is no previous research investigating the mandibular bone density of the children with cyanotic CHD. Since the quality of children’s mandibular bone mineral density affects their periodontal health in adulthood, it is very important to proactively take protective measures and raise awareness. Regarding these measures, both improving the children’s oral hygiene and taking additional precautions for increasing their bone quality ensure periodontal health. Thus, in these patients, periodontal problems which may also adversely affect their cardiac health can be avoided.

## 5. Conclusions

The present study was conducted using the fractal analysis method and panoramic radiomorphometric indices and revealed no significant difference between the healthy children and patients with CHD regarding the mandibular bone trabecular structure and density. However, children in the growth and development stage still have the risk of experiencing the negative effects of CHD in the trabecular structure of bone tissue. In order to reveal the effects of CHD on bone structure, comprehensive studies involving more patients in different age groups are recommended to be carried out.

## Figures and Tables

**Figure 1 children-10-00227-f001:**
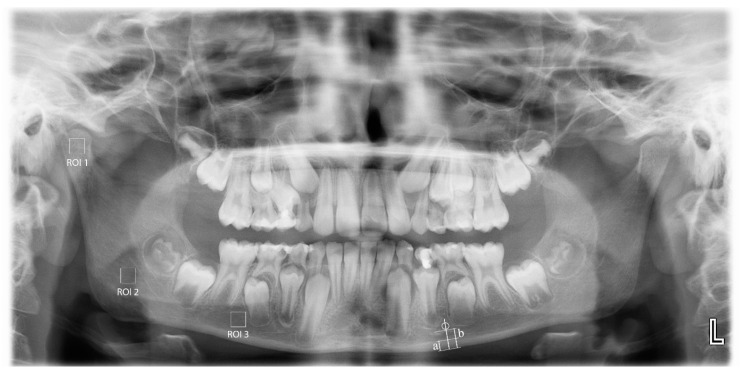
Locations of the regions of interest (ROI) selected from the right mandibular condyle (ROI1), angulus (ROI2), and corpus (ROI3) regions in the panoramic radiographs. Mandibular cortical width (a); distance between the inferior borders of the mental foramen and the mandible (b); and panoramic mandibular index (a/b).

**Figure 2 children-10-00227-f002:**
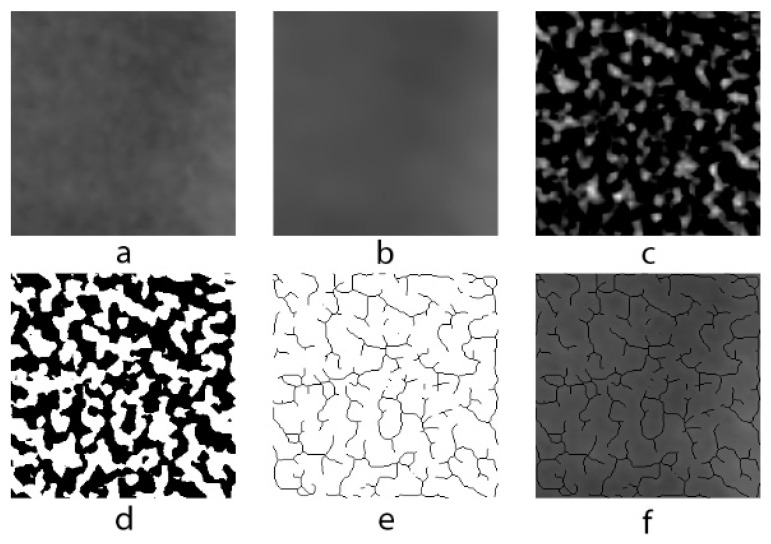
Image processing procedures. (**a**) Raw image before processing. (**b**) Gaussian blurred image. (**c**) Result of subtracting blurred image from raw image. (**d**) Binary image. (**e**) Skeletonized image after processing. (**f**) Composition image of raw and skeletonized image.

**Figure 3 children-10-00227-f003:**
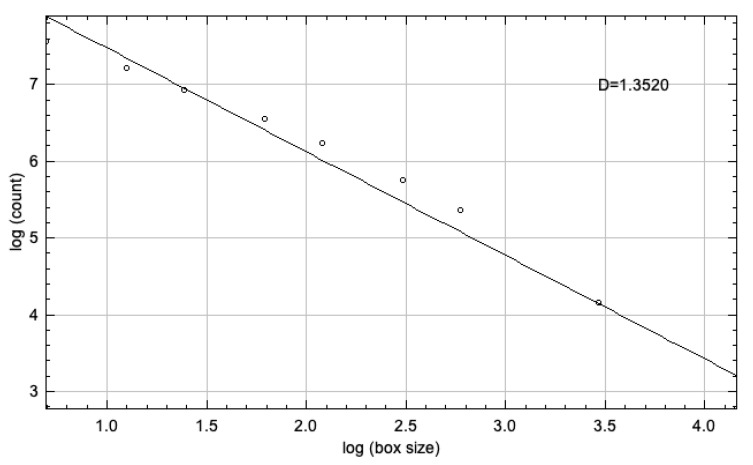
Fractal dimension calculation of the images with box counting method.

**Figure 4 children-10-00227-f004:**
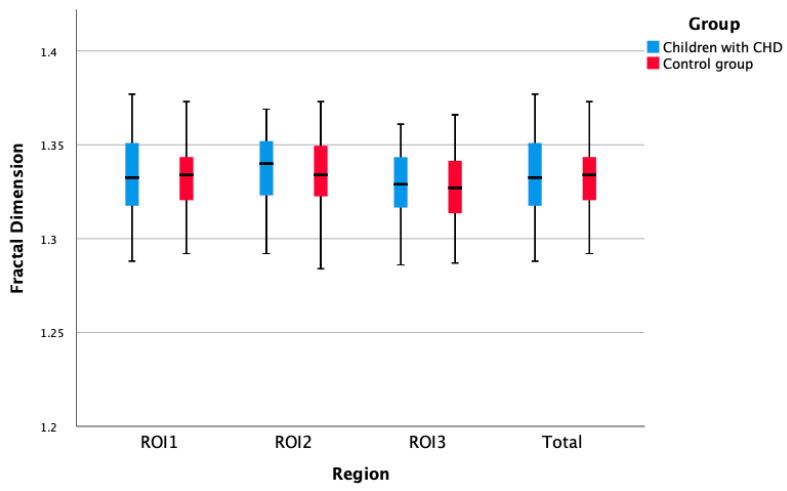
Box-plot representation of fractal dimensions of children with CHD and control group.

**Figure 5 children-10-00227-f005:**
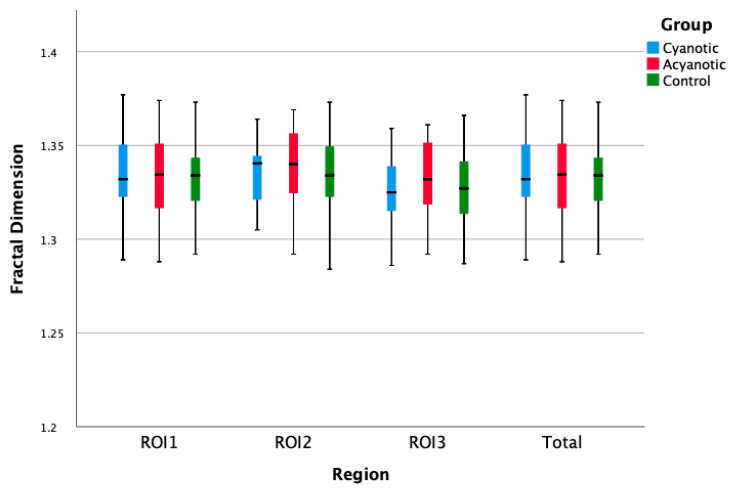
Box-plot representation of fractal measurements of children with CHD.

**Table 1 children-10-00227-t001:** The minimum, maximum, mean, standard deviation, and *p* values of ROI1, ROI2, ROI3, FD, MCW, and PMI in children with CHD and control group.

		Minimum	Maximum	Mean	Standard Deviation	*p*
MCW	Children with CHD	2.13	6.13	3.57	0.86	0.367
	Control	2.14	5.75	3.52	0.88	
	Cyanotic	2.14	5.55	3.51	0.83	0.844
	Acyanotic	2.32	6.06	3.63	0.91	
	Control	2.1	5.75	3.49	0.89	
PMI	Children with CHD	0.23	0.77	0.415	0.111	0.5
	Control	0.26	0.6	0.433	0.078	
	Cyanotic	0.23	0.77	0.437	0.144	0.317
	Acyanotic	0.31	0.5	0.394	0.059	
	Control	0.26	0.6	0.429	0.082	
ROI1	Children with CHD	1.288	1.377	1.334	0.023	0.736
	Control	1.292	1.373	1.331	0.019	
	Cyanotic	1.289	1.377	1.335	0.021	0.882
	Acyanotic	1.288	1.374	1.333	0.025	
	Control	1.292	1.373	1.332	0.019	
ROI2	Children with CHD	1.292	1.369	1.338	0.018	0.381
	Control	1.277	1.373	1.334	0.022	
	Cyanotic	1.305	1.364	1.336	0.017	0.578
	Acyanotic	1.292	1.369	1.34	0.02	
	Control	1.277	1.373	1.334	0.021	
ROI3	Children with CHD	1.286	1.361	1.329	0.02	0.623
	Control	1.287	1.366	1.326	0.019	
	Cyanotic	1.286	1.359	1.325	0.019	0.463
	Acyanotic	1.292	1.361	1.332	0.02	
	Control	1.287	1.366	1.327	0.019	
FD	Children with CHD	1.306	1.361	1.333	0.015	0.53
	Control	1.297	1.362	1.331	0.015	
	Cyanotic	1.306	1.359	1.332	0.014	0.639
	Acyanotic	1.307	1.361	1.335	0.016	
	Control	1.297	1.362	1.331	0.015	

MCW: Mandibular cortical width, PMI: Panoramic mandibular index, FD: Fractal dimension, CHD: Congenital heart disease. Region of interest obtained from the geometric center of the right mandibular condyle (ROI1), just below the mandibular canal in the angelus region of the right mandible (ROI2), and from the apical level of the right teeth 5 and 6 (ROI3).

**Table 2 children-10-00227-t002:** Distribution of MCI and SVE in all groups.

	MCI				SVE		
	C1	C2	C3	*p*	Thin	Not Thin	*p*
Children with CHD	24	5	11	0.222	11	29	0.312
Control	25	1	14		8	32	
Cyanotic	9	4	7	0.083	3	17	0.131
Acyanotic	15	1	4		8	12	
Control	25	1	14		8	32	

CHD: Congenital heart disease, SVE: Simple visual estimation, MCI: Mandibular cortical index (C1: endosteal margins of the cortex are equal and sharp on the right and left sides, C2: endosteal margins show defects in the form of semilunar (lacunar resorption) and/or endosteal cortical residues on one or both sides, C3: the cortical layer is clearly porous and contains heavy endosteal cortical residues).

## Data Availability

The data that support the findings of this study are available from the corresponding author upon reasonable request.

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
