# Peer review of "Morphologic Mandibular Bone Changes on Panoramic Radiographs of Children and Adolescents with Congenital Heart Disease"

_children, 2023, doi:10.3390/children10020227_

Round 1
Reviewer 1 Report
Title Con-genital Rimove dash
Panaromic ... panoramic
Line 30 CHD,40 con-trol) Add a space after the comma, remove dash
Methods
Who performed the radiomorphometric values?
Author Response
Title
Con-genital Rimove dash
-corrected
Panaromic ... panoramic
-corrected
Line 30 CHD,40 con-trol) Add a space after the comma, remove dash
-corrected
Methods
Who performed the radiomorphometric values?
- All panoramic radiographic measurements were performed by a single trained observer (FS), using the ImageJ 1.3 (National Institutes of Health, Bethesda, MD).

Reviewer 2 Report
Specific Comments
Title
Not matching your aim. Please modify.
Please correct the words “Con-genital” and “Panaromic”.
Abstract
The abstract should follow the style of structured abstracts, but without headings.
Line 27: The purpose of the present study “was not is”.
Line 31: FD need to be in full term first then followed by its abbreviation.
In the method part, please add method of statistical analysis. Also, add real P-values.
Please elaborate more on your results. P-value not p-value.
Line 38: Please correct “mandibula”.
Introduction
Line 75: The aim of this retrospective study “was not is”.
All of your mentioned studies focus on children only and not adolescents.
The Introduction does not give a rationale why this study should be conducted or what scientific value it has.
At the end of the Intro section, please give your null hypothesis. The latter should be derived from the preceding thoughts in this section and should be broached again in the Discussion. In hypothesis testing, the null hypothesis is the one you are hoping that is can be disproven by the observed data.
Materials and Methods
English language editing is required.
What is the study rationale?
Please mention sample size calculation.
Please mention study design.
Please add subtitle inclusion criteria.
Please add abbreviation following full term when possible. The first time an abbreviation appears, it should be placed in parentheses following the full spelling of the term.
Line 123: PR need to be in full term first then followed by its abbreviation.
Please add the name of the manufacturer for all materials, X rays and equipment used.
Line 175: Please correct “species”.
Line 194: You mentioned inter-observer variability, what about intra-observer variability.
P-value not p-value.
Results
Please mention kappa coefficient value.
English language editing is required.
Please add real P-values. Please write P-value in italics (P-value).
Line 242: No need for full term regarding “FD”.
Tables and Figures
In Tables 1 and 2, a footnote explaining the abbreviations (ROI1, 2, 3 and C1, C2, C3) needs to be added (what do they stands for). Also, level of significance needs to be mentioned.
Discussion
This section may usefully start with a summary of the major findings, but repetition of parts of the abstract or of the results section should be avoided.
The first paragraph belongs to the introduction section.
Please mention future directions.
Please point out the implications of the findings and their limitations.
Conclusions:
Missing. Please add conclusions subtitle.
References
Also, please check journal guidelines for reference writing.
References needs to be 10 years back not more (from 2012 to 2022).
Old references need to be replaced by recent ones.
All of the journal names need to be abbreviated.
Some of the references include DOI, others do not include DOI number.
In general, all references need to be revised, standardized and written according to the journal guidelines.
Author Response
Title
Not matching your aim. Please modify.
- Morphologic Mandibular Bone Changes on Panoramic Radiographs of Children and Adolescents with Congenital Heart Disease
Please correct the words “Con-genital” and “Panaromic”.
-corrected
Abstract
The abstract should follow the style of structured abstracts, but without headings.
-headings removed from the abstract
Line 27: The purpose of the present study “was not is”.
-corrected
Line 31: FD need to be in full term first then followed by its abbreviation.
-fractal dimension added
In the method part, please add method of statistical analysis. Also, add real P-values.
- Due to the word limit in the abstract, the statistical parts are written short.
Please elaborate more on your results.
-unfortunately due to the word limit in the abstract, the result section was written in short.
P-value not p-value.
-changed to italic.
Line 38: Please correct “mandibula”.
- mandibular bone of
Introduction
Line 75: The aim of this retrospective study “was not is”.
-done
All of your mentioned studies focus on children only and not adolescents.
-added
The Introduction does not give a rationale why this study should be conducted or what scientific value it has.
-added “Dental practitioners should be extremely careful when invasive dental procedures like tooth extraction or traumatic injury management are performed in the patients with low BMD for avoiding treatment complications.”
At the end of the Intro section, please give your null hypothesis. The latter should be derived from the preceding thoughts in this section and should be broached again in the Discussion. In hypothesis testing, the null hypothesis is the one you are hoping that is can be disproven by the observed data.
- The null hypothesis of the study is that there is no difference in the evaluation of cortical and trabecular mandibular bone morphology with fractal analysis and 4 radiomorphometric indices in children with CHD and healthy.
Materials and Methods
English language editing is required.
-checked
What is the study rationale?
- Best to our knowledge, the studies regarding the morphological variations in the mandibulas of the children and adolescents with CHD seem to be unavailable in the literature.
Please mention sample size calculation.
- Based on the Yagmur et al.’s study19 on FD and radiometric indices, the number of patients recruited for this study was sufficient to yield a power of at least 95% for an effect size of 1.2. Calculations are based on standard unpaired t-tests, with two-tailed significance of α = 0.05.
Please mention study design.
-added as retrospective study
Please add subtitle inclusion criteria.
-patient selection title changed as inclusion criteria
Please add abbreviation following full term when possible. The first time an abbreviation appears, it should be placed in parentheses following the full spelling of the term.
-checked
Line 123: PR need to be in full term first then followed by its abbreviation.
-PR changed as panoramic radiograph
Please add the name of the manufacturer for all materials, X rays and equipment used.
- panoramic device (ProMax®, Planmeca Oy, 00880 Helsinki, Finland).
- pulse oximeter (NellcorTM Bedside SpO2 Patient Monitoring System, Medtronic, Minneapolis, MN, USA )
Line 175: Please correct “species”.
-Corrected as spaces
Line 194: You mentioned inter-observer variability, what about intra-observer variability.
- Since the analyzes are performed by only one person, intra-observer analysis cannot be performed.
P-value not p-value.
-p values written in italic
Results
Please mention kappa coefficient value.
- The ICC indicated an excellent reliability for MCW, MCI, SVE, ROI1, and ROI2 (kappa coefficients varied between 0.92 and 0.97) and good reliability for PMI and ROI3 (kappa coefficients were 0.86 and 0.88, respectively).
English language editing is required.
- checked
Please add real P-values. Please write P-value in italics (P-value).
- real p values added in italic
Line 242: No need for full term regarding “FD”.
-full term removed
Tables and Figures
In Tables 1 and 2, a footnote explaining the abbreviations (ROI1, 2, 3 and C1, C2, C3) needs to be added (what do they stands for). Also, level of significance needs to be mentioned.
- footnote explaining the abbreviations were added. Statistically significant difference is presented in the tables in the form of explanation of p values, which are usually marked with asterix (“*”). Since there is no statistically significant difference in our results in the tables, it is not appropriate to explain this situation by associating it with any symbol.
Discussion
This section may usefully start with a summary of the major findings, but repetition of parts of the abstract or of the results section should be avoided.
- Panoramic radiomorphometric indices and FD have been used to determine the effects of many systemic diseases on the mandibula.[4,17,18] Best to our knowledge, the studies regarding the morphological variations in the mandibulas of the children and adolescents with CHD seem to be unavailable in the literature. The present study (depending on whether the children with CHD were cyanotic or acyanotic) was conducted to determine the changes in the mandibulas of the children and adolescents with CHD through FA and dental panoramic radiomorphometric indices (MCW, PMI, MCI, SVE) and it revealed no statistically significant difference regarding trabecular structure and density of the mandibular bone.
The first paragraph belongs to the introduction section.
-partially removed and corrected
Please mention future directions.
- There are some recently conducted studies assessing the bone morphologies of the patients with various systemic diseases with different methods of radiological analysis. However, there is no previous research investigating the mandibular bone density of the children with cyanotic CHD. Since the quality of children’s mandibular bone mineral density affects their periodontal health in adulthood, it is very important to proactively take protective measures and raise awareness. Regarding these measures, both improving the children’s oral hygiene and taking additional precautions for increasing their bone quality ensure the periodontal health. Thus, in these patients, periodontal problems which may also adversely affect their cardiac health can be avoided.
Please point out the implications of the findings and their limitations.
- The limitations of this retrospective study, may be the lack of the occlusal situations that can be affect the mandibular bone morphology and the non-use of DEXA, which is a golden standard in the evaluation of BMD. However, it can contribute to the literature since currently there are no researches assessing the mandibular bone density in the children and adolescents with CHD.
The study samples also included patients who had major operations in the early period into children in the cyanotic CHD group, which can be observed in a small number of patients with the diagnosis and treatment methods that are still under progress in our country. It may have affected the results of the study as it the patients who had major operations in the early period were less exposed to the effects of CHD. In the current study, no evaluation was realized regarding some possible factors for low mandibular bone density, such as physical activity restrictions and other biomarkers that are common in the children affected with CHD. Although this study indicated no difference between the children with CHD and healthy ones in the trabecular mandible morphology, it was recommended to cover these candidate factors in the future studies. It is expected to follow the same patients and compare them with their peers in the adulthood in the further studies.
Conclusions:
Missing. Please add conclusions subtitle.
- Subtitle added
References
Also, please check journal guidelines for reference writing.
-corrected
References needs to be 10 years back not more (from 2012 to 2022).
-all corrected except two
Old references need to be replaced by recent ones.
-replaced
All of the journal names need to be abbreviated.
-corrected
Some of the references include DOI, others do not include DOI number.
-all corrected except one
In general, all references need to be revised, standardized and written according to the journal guidelines.
-done according to the endnote MDPI style

Round 2
Reviewer 2 Report
None. Thank you